behaviour

social contagion, visual contagion, affiliation, behavioural synchrony, tolerance, macaques

**Author for correspondence:**
Julia Ostner
e-mail: julia.ostner@biologie.uni-goettingen.de

# Social contagion of affiliation in female macaques

Julia Ostner[1,2,3], Jana Wilken[1] and Oliver Schülke[1,2,3]

[1]Department Behavioral Ecology, University of Goettingen, Goettingen, Germany
[2]Primate Social Evolution Group, German Primate Centre, Leibniz Institute for Primate Research, Goettingen, Germany
[3]Leibniz ScienceCampus Primate Cognition, Goettingen, Germany

JO, 0000-0001-6871-9976; OS, 0000-0003-0028-9425

Social contagion of non-interactive behaviour is widespread among animals including humans. It is thought to facilitate behavioural synchronization and consequently group cohesion, coordination and opportunities for social learning. Contagion of interactive behaviour—particularly affiliation—has received much less attention. Here, we investigated in female rhesus macaques (*Macaca mulatta*) the effect of observing group members groom on a subject's subsequent grooming behaviour and the potential modulation of contagion by relationship quality and social status. We recorded behaviour after subjects witnessed a grooming event and compared it to behaviour in a control condition with the same individuals in proximity but in the absence of a stimulus grooming event. Compared to the control condition, after observing others groom, females engaged in a grooming interaction sooner, and were more likely to be the initiator and to take on the active groomer role. Dominance rank of the focal individual and more weakly also of the stimulus individuals affected the latency to the next grooming interaction of the focal subject. Latency to the next grooming interaction decreased with increasing rank of the subject potentially reflecting lower social constraints faced by high ranking individuals in this highly despotic species. Relationship quality between the subject and the stimulus individuals had no effect on latency to grooming. Collectively, our findings provide evidence for visual contagion of affiliation in rhesus macaques. Future studies should explore the systematic variation in contagion of interactive behaviour in relation to a gradient of social tolerance.

## 1. Introduction

Social contagion, i.e. the spread of affect or behaviour from one individual to another [1,2], has been documented in a wide range of species, including humans [3]. Functionally, social contagion of behaviour has been proposed to facilitate group

coordination and ultimately group cohesion by activity synchronization [4,5]. In group-living species, behavioural synchronization occurs for example with respect to activity–inactivity transitions [6], vigilance patterns [7,8], and may lead to physiological and affective state-matching [9,10] and increased social bonding [11,12].

The vast majority of studies on behavioural contagion focuses on non-interactive activities, such as contagious yawning and itching/scratching [13,14]. Contagious yawning is probably the best-studied example of non-interactive contagious behaviour and may serve to enhance collective vigilance and to facilitate an adaptive response to external stimuli [15]. It has been documented in a range of highly social species spanning budgerigars [16], rats [17], wolves [18] and several species of primates [19–21], including humans [22]. Yawning contagion has been mostly studied using visual stimuli (for studies on auditory yawning contagion, see [21,23]), either induced by direct observation [18,21] or by attending to video-recordings of yawning conspecifics [16,19,24].

Behavioural contagion of interactive behaviour has received less attention. A few studies on contagion of interactive (e.g. affiliative and aggressive) behaviour in non-human primates provide evidence for an acoustic 'neighbour effect' [25], i.e. the social contagion of behaviour elicited by vocalizations emitted by nearby conspecifics [2]. Captive chimpanzees (*Pan troglodytes*) and common marmosets (*Callithrix jacchus*) when exposed to higher rates of aggressive vocalizations from neighbouring groups, increased their intragroup aggression [2,25,26], while exposure to affiliative neighbour vocalizations elicited increased in-group grooming [2,26]. Similarly, attending to the playback of vocalizations emitted during social play increased rates of play in keas (*Nestor notabilis*; [27]) and of friendly approaches in rats [28].

Visual contagion of social interactions has so far only been shown in one social group of Barbary macaques (*Macaca sylvanus*). In this study, observing group members groom each other promoted affiliative behaviour of a bystanding individual, including her rates of social grooming, contact sitting and friendly approaches [29]. Adding to the limited body of research on contagion of grooming gains relevance as grooming is an integral part of social life in non-human primates. Social grooming represents the main affiliative behavioural pattern in primates, is distributed selectively among potential partners, is instrumental in the formation and maintenance of social bonds [30,31] and increases cooperative behaviour and agonistic support [32–34]. Physiologically, engaging in allo-grooming stimulates the release of oxytocin [35] and beta-endorphins [36] and is associated with reduced anxiety [37,38] and a reduction of glucocorticoid levels [39].

The degree of behavioural contagion of non-interactive behaviour is often modulated by relationship quality between the stimulus individual and the bystander. Contagious yawning is enhanced when the triggering individual is a close social partner in geladas (*Theropithecus gelada*), bonobos (*Pan paniscus*) and humans ([40]; for an opposing effect in contagious scratching, see [14,21,41]) or an in-group versus out-group individual in chimpanzees [42]. Yawn contagion is also facilitated when the triggering individual is of high social status, i.e. the dominant sex, in chimpanzees and bonobos [20,40]. Enhanced behavioural contagion as a function of stimulus' dominance rank or relationship strength between the responder and the stimulus individual has been explained by increased emotional attachment and generally increased salience in signals of closely bonded or high status individuals [20,41,43,44]. Studies on behavioural contagion of interactive behaviour have not tested the contagion modulation by relationship quality or dominance rank.

In this observational study on adult female rhesus macaques (*Macaca mulatta*), we add to the so far only published study on visual contagion of affiliation [29] in several ways: (i) we tested the moderating effects of relationship quality and dominance rank, (ii) we controlled for the presence of additional bystanders in the proximity of the focal (bystander) individual, ruling out that it is the mere presence of others that drives contagious responses (social facilitation), and (iii) we tested the generality of the findings by extending from the socially tolerant Barbary macaques to highly despotic rhesus macaques [45].

Specifically, we predicted that observing others groom would (1) be contagious and thus reduce the time to a bystander's next grooming bout compared to a non-grooming matched control with the exact same individuals in close proximity yet in the absence of others grooming. We further predicted that social contagion would be enhanced (2) with increasing dominance rank of the triggering individuals, and (3) due to the increased opportunity of high ranking responders to start their own grooming interactions, also with increasing dominance rank of the focal individual. Similarly, we predicted enhanced contagion (4) with increasing affiliative relationship strength between the bystander and the groomers. Finally, we predicted that observing grooming would increase the likelihood (5) of focal bystanders initiating a grooming bout rather than being the receiver of a grooming invitation, and (6) of focal bystanders being the groomer rather than the groomee.

# 2. Methods

## 2.1. Study subjects and data collection

The study was conducted in the colony of rhesus macaques housed at the German Primate Center Göttingen, Germany. We observed 19 adult females in three social groups, comprising, respectively, four adult females, one adult male, and one infant (group DUN), eight adult females, one adult male, and one juvenile male (group TED), and seven adult females and one adult male without immatures (group DUS). The larger groups (TED, DUS) were each housed in two adjoining 25 m² indoor compartments plus two approximately 22 m² outdoor enclosures, respectively, the smaller group (DUN) in one 25 m² indoor plus approximately 22 m² outdoor enclosure. The monkeys could move freely between indoor and outdoor areas and the larger groups also between the two indoor and two outdoor enclosures, respectively.

We collected behavioural data using a modified version of the well-established post-conflict–matched control (PC–MC) procedure developed originally to study post-conflict behaviour [46]. In the PC–MC protocol, focal animal observations are conducted on the behaviour of individuals after they have been involved in a conflict (PC sample) and are compared to a focal animal observation of the same individual under the same circumstances but in the absence of a conflict having occurred before (MC). In this study, instead of starting the PC protocol after the focal animal was involved in a conflict, we started focal animal observation after the focal female had observed a grooming interaction involving two group members of which at least one partner had to be an adult female (post-grooming (PG) observation). We then followed the focal female for a maximum of 20 min or until she engaged in a grooming interaction herself. In case the focal animal engaged in grooming, we also recorded (i) whether the focal female was the initiator, defined as approaching another individual and either starting a grooming interaction or presenting to be groomed and (ii) whether she was the groomer or groomee in this interaction [29].

A PG observation started (i) when a female entered the room while a grooming bout was ongoing or (ii) when a grooming bout started and an uninvolved female was in the same room. In both cases to qualify for a PG sample, the bystander female had to visually attend to the grooming interaction judged from head orientation and gaze direction. For every PG observation, on the next possible day and matched for time of day, we collected a corresponding matched control (MC) observation on the same focal female using the same protocol only in the absence of any grooming interactions. In captivity, daily routines of cleaning and feeding are very regular, resulting in external synchronization of group-level behaviour. By matching for time of day, we can exclude that external synchronization affected contagion in just the PG but not the MC observation. Before starting the MC observation, the focal female was observed for 5 min to make sure she had not been involved in a social interaction herself that may have influenced her behaviour. Crucially and to mimic the PG situation as closely as possible, we tried to control for the social environment as much as possible: at the start of the MC data collection the individuals that had been grooming in the PG sample had to be within 2 m proximity of the focal female, as had to be all individuals that had been within 1 m of the focal female. All individuals that had been in 2 m proximity during the PG observation were required to be in the same 25 m² room during the MC. As in the PG observation, the focal female was followed for 20 min or until she engaged in a grooming interaction. In case no MC focal observation could be conducted, the corresponding PG observation was discarded. In total, we collected 106 PG–MC pairs (5–8 per focal female).

## 2.2. Data analysis

Dominance hierarchies including all adult group members were established from the outcome of 528 (DUS), 313 (TED) and 57 (DUN) decided dyadic agonistic interactions [47] using the revised I&SI method and analysed using the DOMICALC software [48]. In all three groups hierarchies were significantly linear. As adult group size ranged from 5 to 9 (including the male), we standardized the ordinal ranks to range from 1 to 9 in all groups with all adults spaced evenly between these values to be comparable across the three groups. In all three groups, the adult male was highest ranking and thus occupied rank position 1.

Affiliative relationship strength was calculated for two (DUS, TED) of the three groups from 180 h of continuous focal animal data collected as part of another project. We extracted the dyadic frequency and duration of three behaviours: grooming, contact sitting and time spent in 1 m proximity. In group TED all

six measures were significantly positively correlated in Kendall's row-wise matrix correlations (mean and standard-deviation across all pair-wise correlations of row-wise average tau = 0.50 ± 0.16, 0.0001 < p < 0.042). In group DUS, the time spent in close proximity was not correlated to the frequency of contact sitting and grooming and the duration of grooming, but all other metrics were significantly positively correlated (row-wise average tau = 0.55 ± 0.16, 0.0001 < p < 0.0125). We also checked that the mean dyadic raw frequency of grooming, contact sitting and being in close proximity was large enough so as to reliably measure variation across dyads (DUS/TED, respectively: proximity 57.0/46.1, contact 26.6/14.1, grooming 23.1/14.8). The correlated metrics were integrated into a dyadic composite sociality index DSI [49]; after controlling all dyadic values for the respective dyadic focal animal observation time, they were divided by the average of that metric across all dyads of the group and then averaged across all metrics to give the DSI of a dyad. The DSI has a mean of 1 and increasingly high values indicate that dyads had been increasingly more often in affiliative contact for increasingly more time than the average dyad in the group, while a dyad with low values spends less than average time affiliating.

Following de Waal & Yoshihara's [46] method, a PG–MC pair was considered 'attracted' if grooming occurred earlier in the PG than the MC observation, 'dispersed' if grooming occurred earlier in MC than the PG observation and 'neutral' if grooming occurred at the same time in both observations or did not occur in either observation. In case a PG or MC did not include any grooming interaction by the focal individual, we set the latency to the next grooming to the maximum observation time, i.e. 1200 s. To avoid pseudo-replication, for each focal female we calculated the proportion of 'attracted' and 'neutral', 'initiated' and 'not-initiated' or 'groomer' and 'groomee' across her respective PG and MC observations and used this individual level value if not using multivariate tests.

Furthermore, we ran the following statistical tests in Statistica 13.3 (TIBCO Software, Inc.). To investigate whether grooming was contagious, we used a Wilcoxon matched-pairs test to test whether an individual's proportion of 'attracted' PG–MC pairs was higher than that of 'dispersed' PG–MC pairs ($N = 19$ focal females: prediction 1; note that the proportion of 'attracted' and 'dispersed' pairs, respectively, is calculated based on the total number of PG–MC pairs, i.e. including the 'neutral' ones). We used a GLM to determine whether the effect of grooming contagion increased with increasing dominance rank of the stimulus groomers (prediction 2) and with increasing rank of the focal (bystander) individual (prediction 3). To do so, we first calculated the difference in latency (in seconds) to the next grooming interaction of the focal animal between the matched control and the corresponding PG-observation (latency MC [s] minus latency PG [s]). We then built a model with this delta latency (in seconds) as the response that increases with increasing contagion. Predictors were the standardized rank of the stimulus individual (selecting the higher of the two) and the standardized rank of the focal subject. Focal subject ID was a random factor ($N = 106$ observations). To test prediction 4 with data from groups DUS and TED, we again used a GLM to determine whether the effect of grooming contagion increased with increasing affiliative relationship strength between the bystander and the groomers. To do so, we built a model with the delta latency (MC minus PG, as above) to the next grooming interaction of the focal animal as the response. The sole test predictor was the DSI value between the focal bystander and the stimulus individuals (selecting the higher of the two). Focal subject ID was a random factor ($N = 82$ observations). Testing prediction 5, we used a Wilcoxon matched-pairs test to assess whether the proportion of next grooming interactions (out of the total number of grooming interactions of the bystander in either condition) for which the bystander was the initiator was higher in PG grooming bouts than in MC grooming bouts. The proportion was calculated based on the total number of grooming events of the focal bystander. Following Berthier & Semple [29], we used only those females for which we had data of at least three grooming interactions in both PG and MC, respectively, which led to the exclusion of six females. For these we had enough (more than 3 grooming interactions) interactions during PG, yet only 0–2 grooming interactions in MC observations. This led to a resulting sample size for this prediction of 13 focal females with an average of 5 (range: 3–8) interactions in PG and an average of 4 (range: 3–6) interactions in MC observations. Finally, testing prediction 6, we again used a Wilcoxon matched-pairs test to determine whether the proportion of an individual's grooming bouts where the bystander was the groomer (in contrast to being groomee) was higher in a PG compared to an MC grooming interaction. Again, the proportion was calculated based on the total number of grooming events of the focal bystander and following Berthier & Semple [29], we used only those females for which we had data of at least three grooming interactions in both PG and MC, again leading to the exclusion of six females, resulting in a sample size of $N = 13$ individuals.

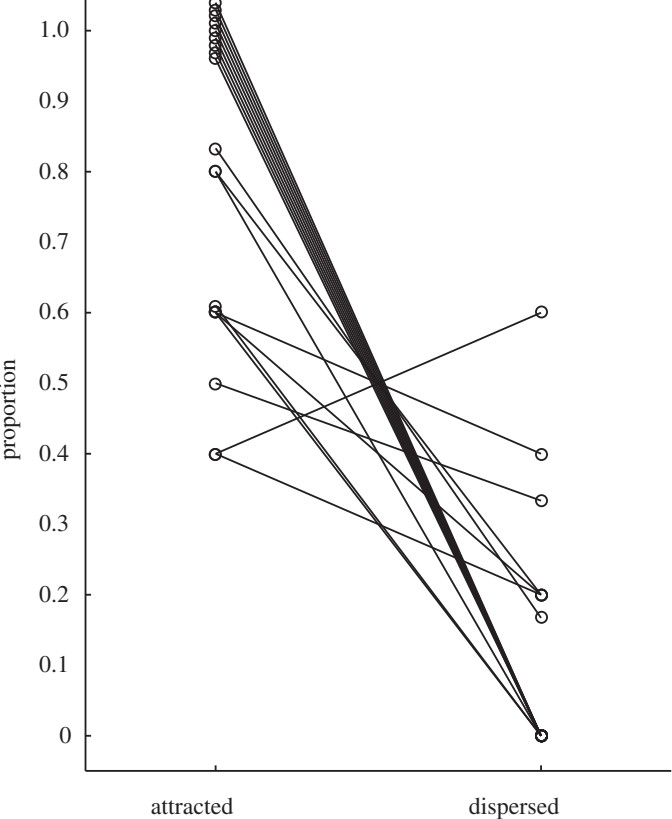

**Figure 1.** Proportion of 'attracted' and 'dispersed' PG–MC pairs for a given focal bystander. Lines connect corresponding values for 19 focal females. Values per female do not necessarily add up to 100 as PG–MC pairs can also be 'neutral'.

## 3. Results

Out of the 106 PG–MC pairs 86 were 'attracted', 11 'dispersed' and 9 'neutral'. Consistent with prediction 1, observing others groom reduced the time to a bystander's next grooming bout compared to a non-grooming matched control at the same time of day and with the same individuals present. Across individuals, the average proportion of 'attracted' PG–MC pairs was significantly higher than the average proportion of 'dispersed' pairs ('attracted': median = 0.83, range 0.40–1.00; 'dispersed': median = 0.00, range = 0.00–0.60; Wilcoxon matched-pairs test: $N = 19$, $Z = 3.7$, $p < 0.001$; figure 1). Only considering PG or MC observation sessions including a grooming interaction of the bystander focal animal (thus omitting observation without focal grooming), the mean latency to a bystander's first grooming interaction was 167 s (range 3–841 s) in a PG and 524 s (range 87–1117 s) in an MC observation. Including all observations, with latencies set to 1200 s if no grooming occurred, the mean latency was 284 s (range 3–1200 s) in a PG and 824 s (range 87–1200 s) in an MC observation.

The dominance rank model was significantly different from the null model with only the random effect (full model: adj. $R^2 = 0.11$, $F = 1.72$, $p = 0.049$; $N = 106$). Both predictors had relatively weak effects on grooming contagion; the higher the dominance rank of the higher ranking of the stimulus individuals the larger was the difference in latency between MC and PG, i.e. the shorter the latency in the PG observation after accounting for latency in the MC (standardized beta = $-0.31 \pm 0.15$, $t = -2.12$, $p = 0.037$; figure 2). Increasing dominance rank of the focal female had a slightly stronger but less significant negative effect on latency to groom, thus the higher ranking an individual the faster she engaged in a grooming interaction herself after observing others groom (standardized beta = $-0.47 \pm 0.25$, $t = -1.87$, $p = 0.065$).

Relationship strength between the focal female and the individuals she witnessed grooming did not influence the degree of contagion; the relationship model was not different from the null model (full model: adj. $R^2 = 0.06$, $F = 1.37$, $p = 0.19$, $N = 82$).

As predicted (prediction 5), the proportion of grooming bouts initiated by the bystander was significantly higher in PG observations, i.e. after observing grooming, compared to MC observations

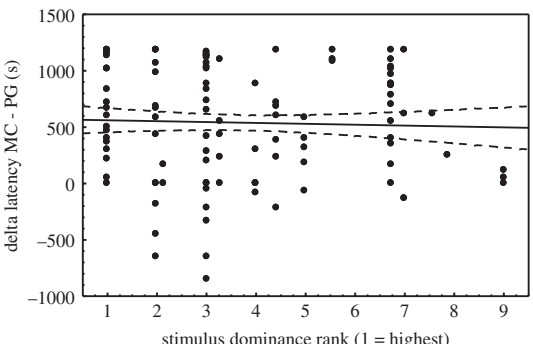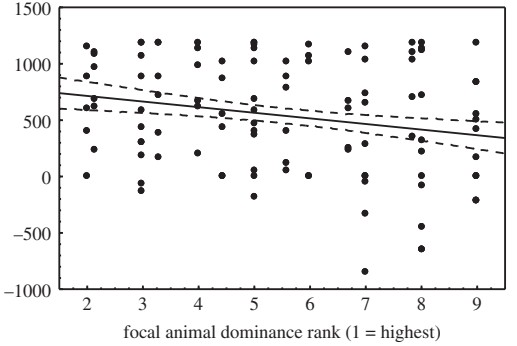

**Figure 2.** Dominance rank effects of stimulus and focal female on the difference in latency between PG and MC observations (large positive differences indicate a much shorter latency in PG, i.e. stronger contagion). Note that dominance ranks were standardized across the groups to account for differences in group size (see methods) and include the focal females as well as the adult male, the latter ranking highest (rank 1) in all groups.

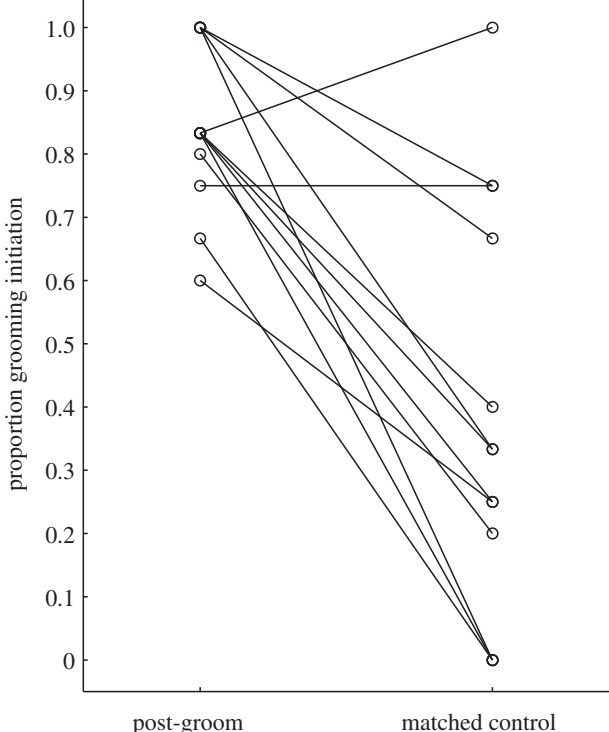

**Figure 3.** Proportion of grooming interactions initiated by the focal individual in post-grooming versus matched control observations, respectively ($N = 13$).

(Wilcoxon matched pairs test: $N = 13$, $Z = 2.98$; $p < 0.01$; figure 3). The average proportion initiated increased from 0.38 in the MC condition to 0.85 in the post-grooming condition. Similarly, observing others groom tended to increase the likelihood of bystanders to take the active role in the next grooming bout. There was a strong statistical trend for a significantly higher proportion of grooming interactions in which the bystander acted as groomer in the PG compared to MC observations (Wilcoxon matched pairs test: $N = 13$, $Z = 1.88$; $p < 0.06$; figure 4).

## 4. Discussion

Our study provides evidence for social contagion of affiliation in rhesus macaques. More than 80% of PG–MC pairs were attracted, suggesting that observing group members groom increased a female's propensity, i.e. lowered her latency, to engage in a grooming interaction herself. Social contagion of

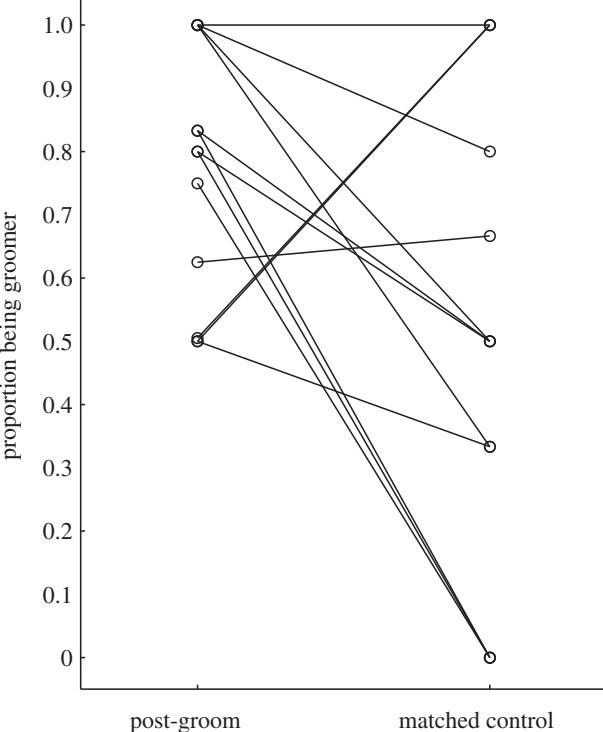

**Figure 4.** Proportion of grooming interactions for which the focal individual was the groomer in PG and MC observations ($N = 13$).

interactive behaviour via the acoustic or—as in our study—the visual domain has been found in a handful of studies so far (see Introduction). Our results most closely resemble findings of visual contagion of grooming behaviour in another macaque species, i.e. the socially more tolerant Barbary macaque [29]. In this and our study, after observing group members groom, females engaged sooner in a grooming interaction themselves and were more likely to initiate the next grooming interaction and to take over the active groomer role (rather than being groomed) compared to a control situation lacking the stimulus grooming observation.

Extending research on social contagion of interactive behaviour, here we investigated the potential moderating effects of dominance rank and relationship quality on contagion propensity found for non-interactive behavioural contagion such as yawning or scratching [14,20,21,40,50]. High ranking or more closely bonded stimulus individuals may constitute stronger cues for social contagion given their overall increased social salience [43,44,51] and also due to the role of contagion for synchrony and group coordination which often is taken on to a large extent by high ranking individuals [20,52]. Despite this conceptual agreement, only few studies investigated the role of dominance rank or relationship quality on contagion tendency. In chimpanzees [20] and bonobos [40], yawning contagion was enhanced when the stimulus individual was of the dominant sex, but dominance rank *per se* was not included in the best model explaining yawning contagion in bonobos [40]. In our study, contagion of grooming behaviour was modulated by stimulus dominance rank, with shorter latencies in the PG condition after witnessing a grooming interaction compared to a control non-grooming condition. This effect, however, was very weak. It has been proposed [20] that dominance effects on contagion probability could be driven by selective attention for dominants [43,51]. And indeed, we have previously shown for the same colony of rhesus macaques that attention is shaped by dominance rank as individuals paid more attention to social interactions of individuals outranking them [44]. In our study here, we ruled out this potential confound by ensuring that focal (bystander) individuals visually attended to the stimulus grooming before we started our post-grooming observation session and by testing against a control condition with the same individuals present.

The latency to the next grooming bout a focal individual engaged in was also affected by the focal individual's own dominance rank: the higher ranking she was, the sooner she herself engaged in a grooming interaction. While this effect was stronger than the effect of the stimulus rank on the latency to groom, the significance of the effect was lower. This result is not simply due to an increased attractiveness of high ranking individuals as grooming partners, as our design directly compared the

two conditions keeping the social environment constant. Instead, increased contagion in high ranking females could be driven by the enhanced social opportunity dominant individuals enjoy. As observing grooming also increased the degree of proactivity in terms of grooming initiation and grooming role (see below), subordinate individuals may simply be more constrained to act on this behavioural tendency.

Societies of female rhesus macaques belong to the most despotic among macaques and exhibit steep hierarchies and minimal social tolerance [45]. While a dominant individual can easily approach any partner and initiate a grooming interaction, a subordinate individual is socially more constrained [53]. Unfortunately, other studies have not incorporated the subject's dominance rank into the analysis, so we cannot investigate interspecific systematic variation of social tolerance and dominance-moderated contagion tendency. Importantly, constraints due to variation in social tolerance should not affect the degree of contagion in non-interactive behaviour, such as yawning, scratching or other solitary behaviours, thus studies contrasting contagion in interactive versus non-interactive behaviour across the gradient of social tolerance would be very informative. Given the range of social styles across macaque species from socially tolerant to highly despotic, this taxon lends itself ideally for such future studies.

Relationship quality, or bondedness, has been shown to enhance yawning contagion in several species, including geladas [21], bonobos [40] and humans [50]. Other studies, however, provide inconsistent evidence, with no effect of relationship quality on yawning contagion in chimpanzees [20] or on grooming contagion as in our study, and an opposing effect for contagion of scratching behaviour in orangutans (*Pongo pygmaeus*), where contagion was more likely to occur if the relationship quality between the stimulus and the observer was low [14]. The lack of an effect of bond strength on contagion in our study could be due to a ceiling effect as contagion degree was strong and latency to the next grooming bout was rather short (284 s on average). Nonetheless, there are several explanations for the proposed link between contagion and bondedness, one building on the prediction that empathy is the underlying proximate mechanism driving yawning contagion [54,55] (but see [56]). A potentially more parsimonious explanation would be that as with dominance (see above) cues from closely bonded individuals are more salient for an observer and thus lead to increased attention [44] and consequently enhanced contagion [56]. As with salience of dominant individuals, we controlled for this potential confound by only investigating 'attended' grooming interactions. If we had not controlled for such a potential attention bias we may have found the predicted relationship, thus we cannot rule out that selective attention for bonded individuals drives enhanced contagion in other studies.

Ultimately, social contagion may enhance behavioural synchrony which in turn facilitates coordination, cohesion and group performance [5,57,58]. Our results suggest that behavioural synchrony may be actively achieved by the bystander. After observing a grooming interaction, females were not only faster to engage in grooming themselves, they also were more proactive to achieve this. In the PG condition, females were far more likely to initiate a grooming interaction and also tended to take the active role of the groomer compared to the control condition. Behavioural synchrony both at the dyad and the group level has been proposed as an adaptive function of social contagion. In humans, positive social contagion and increased synchrony enhances within-group cooperation and overall group performance in experimental set-ups [9,59] and suggestively in less controlled studies [60]. Across social species, behavioural synchrony keeps individuals from drifting apart and thus ensures group cohesion and allows for enhanced within-group cooperation and an increased opportunity for social learning [5,58]. Future studies should investigate whether social contagion and behavioural synchrony indeed comes with a fitness benefit, i.e. whether those individuals, dyads or groups that are better at synchronizing their activities perform better at coordinated activities and ultimately benefit in terms of enhanced survival or reproductive success.

Collectively, our study adds to previous findings highlighting that social interactions do not occur in isolation but in group-living species may also affect the behaviour of others that are not directly involved. Thus, social behaviour needs to be explored beyond the primarily interacting individuals. A dyad or cluster of socially interacting individuals will affect others' social attention to update on the current social situation [44], may lead an observer to interfere by joining or breaking up the situation [61–63] and via social contagion may change observers' propensity to engage in the same type of behaviour themselves.

Ethics. This work followed the Animal Behaviour Society's guidelines for the treatment of animals in behavioural research and teaching, and adhered to the standards as defined by the European Union Council Directive 2010/63/EU on the protection of animals used for scientific purposes. All applicable international, national and/or

institutional guidelines for the care and use of animals were followed. This study was approved by the Ethics Committee of the German Primate Center (AZ E1–19) and was completely observational.

Data accessibility. Data available from the Dryad Digital Repository: https://doi.org/10.5061/dryad.95x69p8hh [64].

Authors' contributions. J.O. and O.S. conceived the study and performed the statistical analyses; J.W. collected the data; J.O. wrote the first draft of the manuscript and all authors contributed to the subsequent drafts. All authors read and approved the final manuscript.

Competing interests. At the time of writing, O.S. is a Board Member of Royal Society Open Science, but had no involvement in the review or assessment of the paper.

Funding. We received no funding for this study.

Acknowledgements. We thank Uwe Schönmann, Annette Husung and the animal keepers at the German Primate Center for logistical support, Anik Brandenburg for providing the DSI dataset and Delphine De Moor for help with data analysis. We thank three anonymous reviewers for excellent comments on the manuscript. The study benefitted from discussions in the Leibniz ScienceCampus Primate Cognition and the DFG-RTG 2070 Understanding Social Relationships (DFG; German Research Foundation, project number 254142454).

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
