## [Reviewer comments · Royal Society Open Science]

Review History

RSOS-201538.R0 (Original submission)

Review form: Reviewer 1

Is the manuscript scientifically sound in its present form?

Yes

Are the interpretations and conclusions justified by the results?

Yes

Is the language acceptable?

Yes

Do you have any ethical concerns with this paper?

No

Have you any concerns about statistical analyses in this paper?

No

Recommendation?

Accept with minor revision (please list in comments)

Comments to the Author(s)

This is an excellent manuscript. The study is well designed and executed and the writing up of the work is clear, logically structured and succinct. The study explore social contagion in rhesus macaques, adding to the very limited body of literature on visual contagion of affiliation. The work improves in a significant ways on the only previous study in the field: through the use of very careful controls in the 'matched-control' context, and in the consideration of relationship quality and dominance rank in mediating contagion. The manuscript was a real pleasure to read and review and makes an important contribution to the field.

I do have two minor issues that I would appreciate the authors addressing.

1. I was a little unclear how rank was calculated here. Was this done for all animals in the group or only for the adult females? And how then was rank allocated? In Figure 2, it states 'rank (alpha=1)' but there are no data points at rank=1 so this makes it appear that there was not an alpha (or was the alpha the adult male, hence missing here?). If the rank of the alpha is 1, I would expect the rank of the beta to be 2, the gamma to be 3 etc. However, figure 2 shows rank values that are (as best I can tell) 2, 2.2, 3, 3.3, 4, 4.4, 5, 5.5, 6, 6.7, 7, 7.8, 8, 9. There appear to be 14 different ranks in the figure, but there are 19 adult females in the analysis. Some clarification on exactly how ranks were assessed, that addresses the issues raised above, would be very helpful!

2. A more minor issue but I think it will help the reader if it is made more clear how PG or MC observations without grooming bouts were dealt with in analyses of latency. I am assuming these were coded as 1200s (the ceiling of the observation period) and think it would make sense to clarify this at the point where it becomes critical. I suggest rephrasing the sentence in lines 161-162 to: "To do so, we built a model with the latency (in seconds) to the next grooming interaction of the focal animal as the response, with latency set to 1200s where no grooming was seen in the observation".

In addition, I found (only!) two trivial errors/typos:

Line 84 - insert 'would' before '(1)'

Line 321 - change 'finding' to 'findings'

Review form: Reviewer 2

Is the manuscript scientifically sound in its present form?

Yes

Are the interpretations and conclusions justified by the results?

Yes

Is the language acceptable?

Yes

Do you have any ethical concerns with this paper?

No

Have you any concerns about statistical analyses in this paper?

No

Recommendation?

Accept with minor revision (please list in comments)

Comments to the Author(s)

This is an interesting study that examines visual contagion of grooming interactions in three groups of captive rhesus macaques. The Authors find that a female bystander observing others being involved in a groom bout displays reduced latency to engage in a grooming bout herself (compared to a non-grooming matched control). The dominance rank of the focal bystander was found to modulate the contagion propensity (higher ranking bystanders had shorter latency before initiating a grooming event), but the relationship quality between the bystander and the highest ranking individual of the observed grooming pair did not influence contagion.

The predictions are clearly presented, the sample size is appropriate, and the statistical framework and testing are well-articulated. I only have minor comments, mostly asking for some clarifications in Methods:

L91-93: I do not understand the difference between prediction 5 (“observing grooming increase the likelihood of focal bystanders of initiating a grooming bout”) and 6 (“observing grooming increase the likelihood of focal bystanders being the groomer rather than the groomee”). Can you clarify? It does not get clearer to me when you describe in more details the calculation and the tests (L172-173 and L176-178). I think that in prediction 5 you only look at the direction of the first act of grooming. While in prediction 6, you take all grooming events into account and calculate the proportion where the focal individual groom others. But you cut your PG and MC observations after the first grooming event (as per L112 and 125). So I am not sure how the proportion differs from the one in prediction 5?

L113: So focal bystanders were only adult females. But could you detail if the grooming bout being observed could be between any age-sex class dyad (i.e. between two adult females, adult male-adult female, juvenile male-adult male or juvenile male-adult female) or only between adults? If the former, I wonder if it would be necessary to control for that, because e.g a grooming between juveniles and females could be less contagious than between two adult females?

L118-120: “Before starting the MC observation, the focal female was observed for five minutes to make sure she had not been involved in a social interaction herself that may have influenced her behavior.” OK I find that 5min is a bit short given that the median latency for contagion is 4.7 min (280s, L189). But I understand the difficulty of the PG-MC design and the fact that the average latency was known after the data collection.

L113: Did you also control for the fact that the focal bystander was not involved in another grooming bout in the previous 5min before starting a PG observation?

L121: “during the MC data collection” could you change by “at the start of the MC data collection” to be more accurate?

L158: For the readers unfamiliar with the PG-MC procedure, maybe clarify that the proportion of attracted and dispersed pairs is calculated based on the total number of PG-MC pairs of an individual (i.e. including the neutral ones).

L161: “We built a model with the latency (in seconds) to the next grooming interaction of the focal animal as the response.”: How did you do when the PG or MC did not include any grooming behaviour? Based on your Figure 2, I think you set the latency at 1200s (i.e. as if it happened at 20min – the end of the observation). Can you clarify in methods.

L172: About the calculation of the “proportion of next grooming interactions for which the bystander was the initiator”: did you calculate this proportion based on the total number of grooming events observed for the focal bystander OR on the total number of PG or MC observations for the focal bystander (in other words, how did you treat the observations without any grooming event). Same question for L177.

L193: “Proportion of ‘attracted’ and ‘dispersed’ PG-MC pairs”: Can you add “for a given focal bystander”?

L199: I am unclear what you report here with the standardized beta: the significance of the simple effect of the focal dominance rank, or of the interaction dominance rank*condition (PG/MC) on the latency? If the former did you estimate it from a model without the interaction? That would be good to add in method which statically test were run to assess the significance of the predictors (LRT test, etc).

L175-176: You excluded 13 or 6 females?

L189: “The mean latency to a bystander’s first grooming interaction was 284s (range 3-1200s)”: Given that your max value is 1200s, I wonder if you include the observations with no grooming event (=set at 1200s) in this calculation. I am not sure if you should. Maybe give first the proportion of PG with a grooming event involving the bystander, and then the true latency without those cases artificially set at 1200s.

L235-240: this feels a bit repetitive with L48-53 in introduction.

Review form: Reviewer 3 (Christine Webb)

Is the manuscript scientifically sound in its present form?

No

Are the interpretations and conclusions justified by the results?

No

Is the language acceptable?

Yes

Do you have any ethical concerns with this paper?

No

Have you any concerns about statistical analyses in this paper?

No

Recommendation?

Reject

Comments to the Author(s)

In this study, researchers assessed whether adult female macaques were more likely to seek grooming interactions having observed other group members engaged in a grooming bout. This is of interest because it widens the scope of social contagion from non-interactive activities (e.g., yawn contagion, which have received the overwhelming focus) to interactive activities like

grooming. I think this is an interesting and understudied idea and thus the paper certainly has merit.

The authors borrowed a methodological approach from the post-conflict literature to test whether grooming was more likely (or faster) to occur after females observed a grooming event (post-grooming: PG) compared to a matched-control (MC) sample in which no such event was observed. Unfortunately, I am not convinced that this approach allowed authors to make systematic comparisons. Specifically, PG and MC samples were not matched on important variables that could have biased their conclusions. I detail these concerns further below but essentially: PG and MC samples were matched by time of day and individuals in proximity but not necessarily the activity of the focal or the non-focal individuals (what was the behavior of the focal and surrounding individuals at the start of the focal?). This left me unconvinced that the lower latencies to groom after watching others groom in the PG window was not an artefact of the context—such as more individuals resting (not walking or feeding) or some other activity that made them more (or less) ‘available’ as grooming partners. I can certainly appreciate that it is a challenge to find corresponding observations which made me wonder whether a better approach would be to develop a method that captures a broader measure (rate) of grooming across all MC observations and compare that to the rate of grooming in the PG window.

Although the manuscript is well-structured and organized, I found that the logic somewhat hard to follow in places, and that the writing occasionally hampered clarity.

1: This title is quite broad given the paper’s focus on adult female rhesus macaques

10-12: Would split this into 2 sentences. “Social contagion of non-interactive behavior is widespread among animals including humans. It is thought to facilitate behavioral synchronization and consequently group cohesion, coordination, and opportunities for social learning.”

17-19: Incomplete sentence; it does not seem like all results / predictions are represented here?

20-22: Please specify direction—i.e., what precise effect did the observer’s dominance rank have?

35-44: I would consider adding a brief statement connecting this elaboration on yawn contagion to the functional implications highlighted in the preceding paragraph—what function is this behavior thought to serve and why is it important / worth elaborating on here? (aside from being a commonly studied and referenced example)

45-48: Unclear immediately why the neighbor effect is an example of interactive behavioral contagion? (you mention yawn contagion, a non-interactive form, can also occur acoustically above)? I guess the assumption here is that the vocalizations involve >1 emitter? i.e., they suggest an ongoing social interaction between two conspecifics? Please clarify.

84-93: As they are currently worded and numbered, the predictions are hard to follow. I might suggest bullet points or some other format. I think it may also be important to allude to your sample demographics here (age-sex class of your subjects), and provide some further justification for some of the predictions. For instance, why would you expect social contagion to increase with the dominance rank of the target and the relationship between the target and observer? Could this be explained by a common mechanism – i.e., more attention towards the target?

115: How was visual attention measured?

116-127: I am a bit concerned about the compatibility between PG and MC samples, though I can appreciate the authors did put thought into this already, and it can be tricky to find an appropriate matched-control.

- 116: Does matched for time of day ensure that animals were engaged in a similar activity – e.g., social activities may occur more/less around feeding or other routine/non-routine management procedures. What the focal doing at the start of the PG/MC observation seems like it may be a more important criteria than time of day.

- 119-120: Wouldn't it also be important to control for whether non-focal individuals in proximity were currently engaged in a grooming bout? Authors specify that all individuals present during the PG observation had to be within 2m of the focal in the MC observation, but not whether their activity was matched (were the non-focal individuals engaged in similar behavior, or explicitly *not* grooming?)

- 120-121: I understand that it is important to control for the social environment but what about the non-social environment; again, the activity of the focal seems crucial here (if the focal was feeding during the MC observation, they would be less likely to begin a grooming bout).

- 121-125: Having tried to conduct MCs like this myself, I feel like this would be very difficult if not impossible (assuming individuals are constantly moving around). Could the authors provide a bit more detail on how they actually managed ensure all of these proximity criteria were met simultaneously (to be clear, I am more impressed than questioning their integrity!)

173-176: 3 grooming interactions in both PG and MC *respectively* or combined across the two samples? What was the mean number of grooming interactions for the remaining females in the sample?

235-240: This is largely redundant with the introduction; in general, I found the discussion rather long. Removing this level of detail on prior studies (perhaps actually adding this back up in the relevant area of the introduction to better set the rationale for the current approach) could help streamline the discussion

262: Not sure what 'gated' means here

Decision letter (RSOS-201538.R0)

Dear Dr Ostner

The Editors assigned to your paper RSOS-201538 "Social contagion of affiliation in macaques" have now received comments from reviewers and would like you to revise the paper in accordance with the reviewer comments and any comments from the Editors. Please note this decision does not guarantee eventual acceptance.

We invite you to respond to the comments supplied below and revise your manuscript. Below the referees' and Editors' comments (where applicable) we provide additional requirements.

Final acceptance of your manuscript is dependent on these requirements being met. We provide guidance below to help you prepare your revision.

Please submit your revised manuscript and required files (see below) no later than 21 days from today's (ie 19-Oct-2020) date. Note: the ScholarOne system will 'lock' if submission of the revision is attempted 21 or more days after the deadline. If you do not think you will be able to meet this deadline please contact the editorial office immediately.

Best regards,

on behalf of Dr Alecia Carter (Associate Editor) and Kevin Padian (Subject Editor)
openscience@royalsociety.org

Associate Editor Comments to Author (Dr Alecia Carter):

I have now received three constructive reviews of your manuscript, and read it myself. I agree with the reviewers that this manuscript is interesting, thorough, well-written, easy to follow, a pleasure to read, and well-executed (but see my comment on the analyses below). In general, the comments are minor and, if addressed, will provide greater clarity in the (small number of!) cases where the writing is not 100% clear.

However, both reviewer 3 and myself had the same concern about the MC condition: from my observations of primates, grooming is more likely to occur when the group is resting, and the MC results could be different if, as reviewer 3 points out, MCs were conducted when resting was the dominant behaviour. (I realise this begs the question – it may well be that ALL behavioural states are contagious and individuals are more likely to rest when others are resting – more on this below.) Do the authors have data on the activity of the group members and the focal individual at the start of the MC? If so, it would be straightforward to analyse whether the behavioural state of the GROUP (e.g. majority resting vs majority feeding / active) predicted shorter latencies to groom, rather than the behavioural state of a PAIR within the group. As reviewer 3 points out, however, it is difficult to design a perfect MC and I would ask that the authors address this

problem explicitly, at least in their discussion, perhaps when they discuss behavioural synchrony at the group level.

I had one last 'large-ish' critique regarding the analyses: Although there is a precedent for the 'simple' categorisation for the analysis for predictions 1, 5 and 6 (L150), this method is 37 years old (!). Are there not more sophisticated and more accurate ways to analyse these data now? It seems odd to make an equivalence between latency differences of e.g. 5 s vs 400 s. Furthermore, a LMM is not appropriate for the other analyses (predictions 2, 3, 4): the latency data are skewed, censored, and represent times-to-events. Wouldn't a survival analysis be more appropriate given these biases? It is possible to perform repeated measures survival analyses. Such an analysis would also address the concerns of reviewer 2 about including the 1200 s latency in some of the calculations and analyses, which, as they point out, biases the measures. If this paper is to 'set a standard' in the field, using a PG-MC approach for grooming contagion, and I believe it will, I would recommend that the more-appropriate survival analyses are used. I realise that this is a pain, and I do not think that the results will change for this study, but if other researchers are to use this article as a guide for the 'appropriate' analyses to do when replicating this approach in future studies, there is greater pressure on this inaugural manuscript to demonstrate a robust statistical approach. I am also confused, for predictions 3, about how the rank of the stimulus individual for the MC condition when no stimulus was observed. Could the authors clarify this?

Minor comment: There should be spaces between numerals and units.

Reviewer comments to Author:

Reviewer: 1

Comments to the Author(s)

This is an excellent manuscript. The study is well designed and executed and the writing up of the work is clear, logically structured and succinct. The study explore social contagion in rhesus macaques, adding to the very limited body of literature on visual contagion of affiliation. The work improves in a significant ways on the only previous study in the field: through the use of very careful controls in the 'matched-control' context, and in the consideration of relationship quality and dominance rank in mediating contagion. The manuscript was a real pleasure to read and review and makes an important contribution to the field.

I do have two minor issues that I would appreciate the authors addressing.

1. I was a little unclear how rank was calculated here. Was this done for all animals in the group or only for the adult females? And how then was rank allocated? In Figure 2, it states 'rank (alpha=1)' but there are no data points at rank=1 so this makes it appear that there was not an alpha (or was the alpha the adult male, hence missing here?). If the rank of the alpha is 1, I would expect the rank of the beta to be 2, the gamma to be 3 etc. However, figure 2 shows rank values that are (as best I can tell) 2, 2.2, 3, 3.3, 4, 4.4, 5, 5.5, 6, 6.7, 7, 7.8, 8, 9. There appear to be 14 different ranks in the figure, but there are 19 adult females in the analysis. Some clarification on exactly how ranks were assessed, that addresses the issues raised above, would be very helpful!

2. A more minor issue but I think it will help the reader if it is made more clear how PG or MC observations without grooming bouts were dealt with in analyses of latency. I am assuming these were coded as 1200s (the ceiling of the observation period) and think it would make sense to clarify this at the point where it becomes critical. I suggest rephrasing the sentence in lines 161-162 to: "To do so, we built a model with the latency (in seconds) to the next grooming interaction of the focal animal as the response, with latency set to 1200s where no grooming was seen in the observation".

In addition, I found (only!) two trivial errors/typos:

Line 84 - insert 'would' before '(1)'

Line 321 - change 'finding' to 'findings'

Reviewer: 2

Comments to the Author(s)

This is an interesting study that examines visual contagion of grooming interactions in three groups of captive rhesus macaques. The Authors find that a female bystander observing others being involved in a groom bout displays reduced latency to engage in a grooming bout herself (compared to a non-grooming matched control). The dominance rank of the focal bystander was found to modulate the contagion propensity (higher ranking bystanders had shorter latency before initiating a grooming event), but the relationship quality between the bystander and the highest ranking individual of the observed grooming pair did not influence contagion.

The predictions are clearly presented, the sample size is appropriate, and the statistical framework and testing are well-articulated. I only have minor comments, mostly asking for some clarifications in Methods:

L91-93: I do not understand the difference between prediction 5 ("observing grooming increase the likelihood of focal bystanders of initiating a grooming bout") and 6 ("observing grooming increase the likelihood of focal bystanders being the groomer rather than the groomee"). Can you clarify? It does not get clearer to me when you describe in more details the calculation and the tests (L172-173 and L176-178). I think that in prediction 5 you only look at the direction of the first act of grooming. While in prediction 6, you take all grooming events into account and calculate the proportion where the focal individual groom others. But you cut your PG and MC observations after the first grooming event (as per L112 and 125). So I am not sure how the proportion differs from the one in prediction 5?

L113: So focal bystanders were only adult females. But could you detail if the grooming bout being observed could be between any age-sex class dyad (i.e. between two adult females, adult male-adult female, juvenile male-adult male or juvenile male-adult female) or only between adults? If the former, I wonder if it would be necessary to control for that, because e.g a grooming between juveniles and females could be less contagious than between two adult females?

L118-120: "Before starting the MC observation, the focal female was observed for five minutes to make sure she had not been involved in a social interaction herself that may have influenced her behavior." OK I find that 5min is a bit short given that the median latency for contagion is 4.7 min (280s, L189). But I understand the difficulty of the PG-MC design and the fact that the average latency was known after the data collection.

L113: Did you also control for the fact that the focal bystander was not involved in another grooming bout in the previous 5min before starting a PG observation?

L121: "during the MC data collection" could you change by "at the start of the MC data collection" to be more accurate?

L158: For the readers unfamiliar with the PG-MC procedure, maybe clarify that the proportion of attracted and dispersed pairs is calculated based on the total number of PG-MC pairs of an individual (i.e. including the neutral ones).

L161: "We built a model with the latency (in seconds) to the next grooming interaction of the focal animal as the response.": How did you do when the PG or MC did not include any grooming behaviour? Based on your Figure 2, I think you set the latency at 1200s (i.e. as if it happened at 20min - the end of the observation). Can you clarify in methods.

L172: About the calculation of the "proportion of next grooming interactions for which the bystander was the initiator": did you calculate this proportion based on the total number of grooming events observed for the focal bystander OR on the total number of PG or MC observations for the focal bystander (in other words, how did you treat the observations without any grooming event). Same question for L177.

L193: "Proportion of 'attracted' and 'dispersed' PG-MC pairs": Can you add "for a given focal bystander"?

L199: I am unclear what you report here with the standardized beta: the significance of the simple effect of the focal dominance rank, or of the interaction dominance rank*condition (PG/MC) on the latency? If the former did you estimate it from a model without the interaction? That would be good to add in method which statically test were run to assess the significance of the predictors (LRT test, etc).

L175-176: You excluded 13 or 6 females?

L189: "The mean latency to a bystander's first grooming interaction was 284s (range 3-1200s)": Given that your max value is 1200s, I wonder if you include the observations with no grooming event (=set at 1200s) in this calculation. I am not sure if you should. Maybe give first the proportion of PG with a grooming event involving the bystander, and then the true latency without those cases artificially set at 1200s.

L235-240: this feels a bit repetitive with L48-53 in introduction.

Reviewer: 3

Comments to the Author(s)

In this study, researchers assessed whether adult female macaques were more likely to seek grooming interactions having observed other group members engaged in a grooming bout. This is of interest because it widens the scope of social contagion from non-interactive activities (e.g., yawn contagion, which have received the overwhelming focus) to interactive activities like grooming. I think this is an interesting and understudied idea and thus the paper certainly has merit.

The authors borrowed a methodological approach from the post-conflict literature to test whether grooming was more likely (or faster) to occur after females observed a grooming event (post-grooming: PG) compared to a matched-control (MC) sample in which no such event was observed. Unfortunately, I am not convinced that this approach allowed authors to make systematic comparisons. Specifically, PG and MC samples were not matched on important variables that could have biased their conclusions. I detail these concerns further below but essentially: PG and MC samples were matched by time of day and individuals in proximity but not necessarily the activity of the focal or the non-focal individuals (what was the behavior of the focal and surrounding individuals at the start of the focal?). This left me unconvinced that the

lower latencies to groom after watching others groom in the PG window was not an artefact of the context – such as more individuals resting (not walking or feeding) or some other activity that made them more (or less) ‘available’ as grooming partners. I can certainly appreciate that it is a challenge to find corresponding observations which made me wonder whether a better approach would be to develop a method that captures a broader measure (rate) of grooming across all MC observations and compare that to the rate of grooming in the PG window.

Although the manuscript is well-structured and organized, I found that the logic somewhat hard to follow in places, and that the writing occasionally hampered clarity.

1: This title is quite broad given the paper’s focus on adult female rhesus macaques

10-12: Would split this into 2 sentences. “Social contagion of non-interactive behavior is widespread among animals including humans. It is thought to facilitate behavioral synchronization and consequently group cohesion, coordination, and opportunities for social learning.”

17-19: Incomplete sentence; it does not seem like all results / predictions are represented here?

20-22: Please specify direction – i.e., what precise effect did the observer’s dominance rank have?

35-44: I would consider adding a brief statement connecting this elaboration on yawn contagion to the functional implications highlighted in the preceding paragraph – what function is this behavior thought to serve and why is it important / worth elaborating on here? (aside from being a commonly studied and referenced example)

45-48: Unclear immediately why the neighbor effect is an example of interactive behavioral contagion? (you mention yawn contagion, a non-interactive form, can also occur acoustically above)? I guess the assumption here is that the vocalizations involve >1 emitter? i.e., they suggest an ongoing social interaction between two conspecifics? Please clarify.

84-93: As they are currently worded and numbered, the predictions are hard to follow. I might suggest bullet points or some other format. I think it may also be important to allude to your sample demographics here (age-sex class of your subjects), and provide some further justification for some of the predictions. For instance, why would you expect social contagion to increase with the dominance rank of the target and the relationship between the target and observer? Could this be explained by a common mechanism – i.e., more attention towards the target?

115: How was visual attention measured?

116-127: I am a bit concerned about the compatibility between PG and MC samples, though I can appreciate the authors did put thought into this already, and it can be tricky to find an appropriate matched-control.

- 116: Does matched for time of day ensure that animals were engaged in a similar activity – e.g., social activities may occur more/less around feeding or other routine/non-routine management procedures. What the focal doing at the start of the PG/MC observation seems like it may be a more important criteria than time of day.

- 119-120: Wouldn’t it also be important to control for whether non-focal individuals in proximity were currently engaged in a grooming bout? Authors specify that all individuals present during the PG observation had to be within 2m of the focal in the MC observation, but

not whether their activity was matched (were the non-focal individuals engaged in similar behavior, or explicitly *not* grooming?)

- 120-121: I understand that it is important to control for the social environment but what about the non-social environment; again, the activity of the focal seems crucial here (if the focal was feeding during the MC observation, they would be less likely to begin a grooming bout).

- 121-125: Having tried to conduct MCs like this myself, I feel like this would be very difficult if not impossible (assuming individuals are constantly moving around). Could the authors provide a bit more detail on how they actually managed ensure all of these proximity criteria were met simultaneously (to be clear, I am more impressed than questioning their integrity!)

173-176: 3 grooming interactions in both PG and MC *respectively* or combined across the two samples? What was the mean number of grooming interactions for the remaining females in the sample?

235-240: This is largely redundant with the introduction; in general, I found the discussion rather long. Removing this level of detail on prior studies (perhaps actually adding this back up in the relevant area of the introduction to better set the rationale for the current approach) could help streamline the discussion

262: Not sure what 'gated' means here

===PREPARING YOUR MANUSCRIPT===

===PREPARING YOUR REVISION IN SCHOLARONE===

Author's Response to Decision Letter for (RSOS-201538.R0)

See Appendix A.

Decision letter (RSOS-201538.R1)

Dear Dr Ostner,

It is a pleasure to accept your manuscript entitled "Social contagion of affiliation in female macaques" in its current form for publication in Royal Society Open Science. The feedback from the Editors is included at the foot of this letter.

on behalf of Dr Alecia Carter (Associate Editor) and Kevin Padian (Subject Editor)
openscience@royalsociety.org

Associate Editor Comments to Author (Dr Alecia Carter):

I am satisfied with the authors' responses to the critiques raised by the reviewers.

Appendix A

Göttingen, 1 December 2020

Dear Dr. Carter, dear reviewers,

Thank you very much for these insightful and constructive comments, which helped to improve our manuscript substantially. We addressed all comments made by the editor and reviewers, most importantly we re-run the models for the dominance and relationship effects on contagion using a different approach and we added detail about the methods throughout the manuscript. We hope the manuscript is now acceptable for publication. Below, we respond point-by-point to the comments raised by the reviewers.

Best regards,

Julia Ostner

Associate Editor Comments to Author (Dr Alecia Carter):

I have now received three constructive reviews of your manuscript, and read it myself. I agree with the reviewers that this manuscript is interesting, thorough, well-written, easy to follow, a pleasure to read, and well-executed (but see my comment on the analyses below). In general, the comments are minor and, if addressed, will provide greater clarity in the (small number of!) cases where the writing is not 100% clear.

Response: Thank you for this overall very positive assessment of our manuscript.

However, both reviewer 3 and myself had the same concern about the MC condition: from my observations of primates, grooming is more likely to occur when the group is resting, and the MC results could be different if, as reviewer 3 points out, MCs were conducted when resting was the dominant behaviour. (I realise this begs the question—it may well be that ALL behavioural states are contagious and individuals are more likely to rest when others are resting—more on this below.) Do the authors have data on the activity of the group members and the focal individual at the start of the MC? If so, it would be straightforward to analyse whether the behavioural state of the GROUP (e.g. majority resting vs majority feeding / active) predicted shorter latencies to groom, rather than the behavioural state of a PAIR within the group. As reviewer 3 points out, however, it is difficult to design a perfect MC and I would ask that the authors address this problem explicitly, at least in their discussion, perhaps when they discuss behavioural synchrony at the group level.

Response: This is a very important concern that has also been raised by reviewer 3. We do not have data on group activity to tests for this effect. Yet we feel this problem is alleviated in our case by the study design and the highly structured captive conditions with a daily recurring routine. By time matching we exclude the danger of systematically selecting a specific group activity in one condition and another

activity in the other condition. We added this in the revised manuscript: *“In captivity daily routines of cleaning and feeding are very regular resulting in external synchronization of group level behavior. By matching for time of day, we can exclude that external synchronization affected contagion in just the PG but not the MC observation.”* (line 129-132)

I had one last ‘large-ish’ critique regarding the analyses: Although there is a precedent for the ‘simple’ categorisation for the analysis for predictions 1, 5 and 6 (L150), this method is 37 years old (!). Are there not more sophisticated and more accurate ways to analyse these data now? It seems odd to make an equivalence between latency differences of e.g. 5 s vs 400 s. Furthermore, a LMM is not appropriate for the other analyses (predictions 2, 3, 4): the latency data are skewed, censored, and represent times-to-events. Wouldn’t a survival analysis be more appropriate given these biases? It is possible to perform repeated measures survival analyses. Such an analysis would also address the concerns of reviewer 2 about including the 1200 s latency in some of the calculations and analyses, which, as they point out, biases the measures. If this paper is to ‘set a standard’ in the field, using a PG-MC approach for grooming contagion, and I believe it will, I would recommend that the more-appropriate survival analyses are used. I realise that this is a pain, and I do not think that the results will change for this study, but if other researchers are to use this article as a guide for the ‘appropriate’ analyses to do when replicating this approach in future studies, there is greater pressure on this inaugural manuscript to demonstrate a robust statistical approach. I am also confused, for predictions 3, about how the rank of the stimulus individual for the MC condition when no stimulus was observed. Could the authors clarify this?

Response: We thank the editor for bringing up these important points:

Concerning the almost 40 year-old method: with these analyses (testing predictions 1,5,6) we did not intend primarily to set a standard, instead we aimed to replicate the recent study by Berthier & Semple 2018 published in Proceedings B that used exactly this method. We thus prefer to stick to these methods when testing these specific predictions.

As to the use of censored data in a LMM, we had overlooked this problem indeed and are thankful for the editor’s attention to this flaw. We spent quite some time searching for survival analysis appropriate for our data as suggested by the editor. There are indeed types of survival analyses available that accommodate repeated sampling in the sense that the time to event is evaluated for the same individual in successive time blocks with time-varying covariates. What is not possible though, is to set the same individual back to zero in the next observation as we would need to do here. Therefore, we settled on a model maintaining the powerful PG-MC approach, but without censored data. Specifically, we calculated the difference in latency (latency MC observation minus latency PG observation) to the next grooming interaction by the focal subject and used this value as the response variable. We kept the censored data, i.e. 1200 s in PG or MC observations without a grooming interaction, but as a result of the subtraction we did not end up with censored data as the response variable (the delta latency never was 1200). These new analyses confirmed the previous results showing an effect of the focal animal’s dominance and no effect of relationship strength on latency to next grooming. In contrast to the previous analysis, rank of the stimulus individual now also has a (weak) significant effect in the predicted direction. We changed the methods, results and discussion to accommodate these changes (lines 180-194; 228 – 245; 284-300).

As to the final comment on the rank of stimulus in the MC condition: the stimulus individual is the same in the PG and the corresponding MC as we made sure the same individuals were around in the control conditions (just in the control the stimulus individual was not grooming). This procedure is hopefully less confusing in the revised version, as now the PG-MC observation remain linked when calculating the delta latency.

Minor comment: There should be spaces between numerals and units.

Response: has been changed throughout

Reviewer comments to Author:

Reviewer: 1

Comments to the Author(s)

This is an excellent manuscript. The study is well designed and executed and the writing up of the work is clear, logically structured and succinct. The study explore social contagion in rhesus macaques, adding to the very limited body of literature on visual contagion of affiliation. The work improves in a significant ways on the only previous study in the field: through the use of very careful controls in the 'matched-control' context, and in the consideration of relationship quality and dominance rank in mediating contagion. The manuscript was a real pleasure to read and review and makes an important contribution to the field.

Response: We are happy the reviewer appreciated our study and the added value to previous work on the topic.

I do have two minor issues that I would appreciate the authors addressing.

1. I was a little unclear how rank was calculated here. Was this done for all animals in the group or only for the adult females? And how then was rank allocated? In Figure 2, it states 'rank (alpha=1)' but there are no data points at rank=1 so this makes it appear that there was not an alpha (or was the alpha the adult male, hence missing here?). If the rank of the alpha is 1, I would expect the rank of the beta to be 2, the gamma to be 3 etc. However, figure 2 shows rank values that are (as best I can tell) 2, 2.2, 3, 3.3, 4, 4.4, 5, 5.5, 6, 6.7, 7, 7.8, 8, 9. There appear to be 14 different ranks in the figure, but there are 19 adult females in the analysis. Some clarification on exactly how ranks were assessed, that addresses the issues raised above, would be very helpful!

Response: sorry for having been unclear in the previous version. We calculated an ordinal dominance hierarchy in each of the three groups including the male and standardized the ranks to match the different group size. We explain this in lines 147 - 150: "*As adult group size ranged from 5 to 9 (including the male), we then standardized the ordinal ranks to range from 1 to 9 with all adults spaced evenly between these values to be comparable across the three groups. In all three groups the adult male was highest ranking and thus occupied rank position 1.*" We also remind the reader again of the standardized rank and the male ranking on the highest position in the legend of Figure 2.

2. A more minor issue but I think it will help the reader if it is made more clear how PG or MC observations without grooming bouts were dealt with in analyses of latency. I am assuming these were

coded as 1200s (the ceiling of the observation period) and think it would make sense to clarify this at the point where it becomes critical. I suggest rephrasing the sentence in lines 161-162 to: "To do so, we built a model with the latency (in seconds) to the next grooming interaction of the focal animal as the response, with latency set to 1200s where no grooming was seen in the observation".

Response: We apologize for not having been clearer about how we dealt with these censored data. Observations without any grooming had in fact been coded as 1200 s. We mention this explicitly now in line 171 - 172 and also calculate mean latency for PG and MC with and without these censored data points (following a comment by Reviewer 2, line 218 - 222): *"Only considering PG or MC observation sessions including a grooming interaction of the bystander focal animal (thus omitting observation without focal grooming), the mean latency to a bystander's first grooming interaction was 167 s (range 3 - 841 s) in a PG and 524 s (range 87 - 1117 s) in a MC observation. Including all observations, with latencies set to 1200 s if no grooming occurred, the mean latency was 284 s (range 3-1200 s) in a PG and 824s (range 87-1200 s) in a MC observation."*

In addition, I found (only!) two trivial errors/typos:

Line 84 - insert 'would' before '(1)'

Response: changed as suggested (line 90)

Line 321 - change 'finding' to 'findings'

Response: changed as suggested (line 349)

Reviewer: 2

Comments to the Author(s)

This is an interesting study that examines visual contagion of grooming interactions in three groups of captive rhesus macaques. The Authors find that a female bystander observing others being involved in a groom bout displays reduced latency to engage in a grooming bout herself (compared to a non-grooming matched control). The dominance rank of the focal bystander was found to modulate the contagion propensity (higher ranking bystanders had shorter latency before initiating a grooming event), but the relationship quality between the bystander and the highest ranking individual of the observed grooming pair did not influence contagion.

The predictions are clearly presented, the sample size is appropriate, and the statistical framework and testing are well-articulated. I only have minor comments, mostly asking for some clarifications in Methods:

Response: thank you for the positive comments on our manuscript

L91-93: I do not understand the difference between prediction 5 ("observing grooming increase the likelihood of focal bystanders of initiating a grooming bout") and 6 ("observing grooming increase the likelihood of focal bystanders being the groomer rather than the groomee"). Can you clarify? It does not get clearer to me when you describe in more details the calculation and the tests (L172-173 and L176-178). I think that in prediction 5 you only look at the direction of the first act of grooming. While in

prediction 6, you take all grooming events into account and calculate the proportion where the focal individual groom others. But you cut your PG and MC observations after the first grooming event (as per L112 and 125). So I am not sure how the proportion differs from the one in prediction 5?

Response: We realize that this distinction was indeed not very clear in the previous version. In our study and following Berthier & Semple (2018), subjects were initiators if they approached another individual and either started a grooming interaction or presented to be groomed. If individual start grooming after individual B approached and presented/invited grooming, B is the initiator and A the active part. Thus, initiation and active grooming are not always the same. This information was added to the revised manuscript (line 119 - 123): *“In case the focal animal engaged in grooming we recorded (i) whether the focal female was the initiator, defined as approaching another individual and either starting a grooming interaction or presenting to be groomed and (ii) whether she was the groomer or groomee in this interaction (Berthier & Semple 2018).”*

L113: So focal bystanders were only adult females. But could you detail if the grooming bout being observed could be between any age-sex class dyad (i.e. between two adult females, adult male-adult female, juvenile male-adult male or juvenile male-adult female) or only between adults? If the former, I wonder if it would be necessary to control for that, because e.g a grooming between juveniles and females could be less contagious than between two adult females?

Response: To be used in our study, a PG grooming interaction had to involve at least one adult female. We added this sentence to the manuscript (line 118). We did not control for the age-sex class of the grooming partner.

L118-120: “Before starting the MC observation, the focal female was observed for five minutes to make sure she had not been involved in a social interaction herself that may have influenced her behavior.” OK I find that 5min is a bit short given that the median latency for contagion is 4.7 min (280s, L189). But I understand the difficulty of the PG-MC design and the fact that the average latency was known after the data collection.

Response: Our pre-MC observation window of 5 min indeed is not very long. If a focal female however had observed a grooming more than five minutes before the start of the MC, this would introduce a conservative error, as this would possibly prompt contagion leading to shortened latencies in the MC.

L113: Did you also control for the fact that the focal bystander was not involved in another grooming bout in the previous 5min before starting a PG observation?

Response: We did not control for a focal animal’s attention or behavior prior to the stimulus grooming observation. We are not sure how that could have affected the interpretation of our results.

L121: “during the MC data collection” could you change by “at the start of the MC data collection” to be more accurate?

Response: changed as suggested (line 135)

L158: For the readers unfamiliar with the PG-MC procedure, maybe clarify that the proportion of attracted and dispersed pairs is calculated based on the total number of PG-MC pairs of an individual (i.e. including the neutral ones).

Response: we added this information as suggested (line 179 - 180): *“note that the proportion of ‘attracted’ and ‘dispersed’ pairs, respectively, is calculated based on the total number of PG/MC pairs, i.e. including the ‘neutral ones.’”*

L161: “We built a model with the latency (in seconds) to the next grooming interaction of the focal animal as the response.”: How did you do when the PG or MC did not include any grooming behaviour? Based on your Figure 2, I think you set the latency at 1200s (i.e. as if it happened at 20min – the end of the observation). Can you clarify in methods.

Response: We indeed set the latency at the maximum observation time, i.e. 1200. We mention this now in the revised version (line 171 - 172): *“In case, a PG or MC did not include any grooming interaction by the focal individual, we set the latency to the next grooming to the maximum observation time, i.e. 1200 s.”*

L172: About the calculation of the “proportion of next grooming interactions for which the bystander was the initiator”: did you calculate this proportion based on the total number of grooming events observed for the focal bystander OR on the total number of PG or MC observations for the focal bystander (in other words, how did you treat the observations without any grooming event). Same question for L177.

Response: Sorry for not having been clear. The proportions ‘grooming initiation’ and ‘active groomer’ were calculated based on the total number of grooming events of the focal bystander during post groom, respectively, matched control observations (not on the total number of PG/MC sessions). We added this information to the text (line 197 - 198): *“The proportion was calculated based on the total number of grooming events of the focal bystander.”*

L193: “Proportion of ‘attracted’ and ‘dispersed’ PG-MC pairs”: Can you add “for a given focal bystander”?

Response: changed accordingly (line 225)

L199: I am unclear what you report here with the standardized beta: the significance of the simple effect of the focal dominance rank, or of the interaction dominance rank*condition (PG/MC) on the latency? If the former did you estimate it from a model without the interaction? That would be good to add in method which statically test were run to assess the significance of the predictors (LRT test, etc).

Response: The interaction term is no longer part of our statistical model. Beta is the model estimate of the regression slope.

L175-176: You excluded 13 or 6 females?

Response: We excluded 6 of the 19 females, leaving a total sample size of 13 for this test. We reworded this sentence to clarify (Line ca 206 - 209): *“Again, the proportion was calculated based on the total number of grooming events of the focal bystander and following Berthier and Semple (2018), we used only those females for which we had data of at least 3 grooming interactions in both PG and MC, again leading to the exclusion of 6 females, resulting in a sample size of N = 13 individuals.”*

L189: “The mean latency to a bystander’s first grooming interaction was 284s (range 3-1200s)”: Given that your max value is 1200s, I wonder if you include the observations with no grooming event (=set at 1200s) in this calculation. I am not sure if you should. Maybe give first the proportion of PG with a grooming

event involving the bystander, and then the true latency without those cases artificially set at 1200s. Response: As suggested, we now report mean and range for latencies with and without censored sessions (latency set to 1200s; line 218 – 222): *“Only considering PG or MC observation sessions including a grooming interaction of the bystander focal animal (thus omitting observation without focal grooming), the mean latency to a bystander’s first grooming interaction was 167 s (range 3 – 841 s) in a PG and 524 s (range 87 – 1117 s) in a MC observation. Including all observations, with latencies set to 1200 s if no grooming occurred, the mean latency was 284s (range 3-1200 s) in a PG and 824s (range 87-1200 s) in a MC observation.”*

L235-240: this feels a bit repetitive with L48-53 in introduction.

Response: we agree and deleted this section in the revised version

Reviewer: 3

Comments to the Author(s)

In this study, researchers assessed whether adult female macaques were more likely to seek grooming interactions having observed other group members engaged in a grooming bout. This is of interest because it widens the scope of social contagion from non-interactive activities (e.g., yawn contagion, which have received the overwhelming focus) to interactive activities like grooming. I think this is an interesting and understudied idea and thus the paper certainly has merit.

Response: Thanks for this comment, we are happy the reviewer sees the merit of this study

The authors borrowed a methodological approach from the post-conflict literature to test whether grooming was more likely (or faster) to occur after females observed a grooming event (post-grooming: PG) compared to a matched-control (MC) sample in which no such event was observed. Unfortunately, I am not convinced that this approach allowed authors to make systematic comparisons. Specifically, PG and MC samples were not matched on important variables that could have biased their conclusions. I detail these concerns further below but essentially: PG and MC samples were matched by time of day and individuals in proximity but not necessarily the activity of the focal or the non-focal individuals (what was the behavior of the focal and surrounding individuals at the start of the focal?). This left me unconvinced that the lower latencies to groom after watching others groom in the PG window was not an artefact of the context—such as more individuals resting (not walking or feeding) or some other activity that made them more (or less) ‘available’ as grooming partners. I can certainly appreciate that it is a challenge to find corresponding observations which made me wonder whether a better approach would be to develop a method that captures a broader measure (rate) of grooming across all MC observations and compare that to the rate of grooming in the PG window.

Response: Thanks for this important comment, that was reiterated by the editor. Given the combination of our time matched PG-MC study design and the captive setting, a systematic bias in group activity between the MC and PG conditions seems highly unlikely. Given the highly structured captive conditions with a daily recurring routine, by time matching we exclude the danger of systematically selecting a specific group activity in one condition and another activity in the other condition. We added this in the revised manuscript (line 129 - 132): *“In captivity daily routines of cleaning and feeding are very regular resulting in external synchronization of group level behavior. By matching for time of day, we can exclude that external synchronization affected contagion in just the PG but not the MC observation.”*

Although the manuscript is well-structured and organized, I found that the logic somewhat hard to follow in places, and that the writing occasionally hampered clarity.

Response: We are sorry to hear this. We double checked the writing and incorporated all comments by the editor and reviewers regarding writing, so we hope the revised version is clearer now.

1: This title is quite broad given the paper's focus on adult female rhesus macaques

Response: We changed the title to read "Social contagion of affiliation in female macaques"

10-12: Would split this into 2 sentences. "Social contagion of non-interactive behavior is widespread among animals including humans. It is thought to facilitate behavioral synchronization and consequently group cohesion, coordination, and opportunities for social learning."

Response: changed as suggested (line 10 – 12)

17-19: Incomplete sentence; it does not seem like all results / predictions are represented here?

Response: We are not entirely sure, how this sentence is incomplete. We changed it slightly to increase readability. Indeed, we do not summarize all results in this one sentence; the results on the moderating effect of dominance rank and relationship quality are summarized in the subsequent sentence. (line 17 – 24):

20-22: Please specify direction—i.e., what precise effect did the observer's dominance rank have?

Response: As the results have changed slightly following the re-analyses, this sentence has been changed. In the revised version we specify the direction of the rank effects (line 21 - 23): "*Latency to the next grooming interaction decreased with increasing rank of the subject potentially reflecting lower social constraints faced by high ranking individuals in this highly despotic species.*"

35-44: I would consider adding a brief statement connecting this elaboration on yawn contagion to the functional implications highlighted in the preceding paragraph—what function is this behavior thought to serve and why is it important / worth elaborating on here? (aside from being a commonly studied and referenced example)

Response: We added a sentence on the proposed functional role of contagious yawning (line 38 – 41): "*Contagious yawning is probably the best-studied example of non-interactive contagious behavior and may serve to enhance collective vigilance and to facilitate an adaptive response to external stimuli (Massen and Gallup 2017).*"

45-48: Unclear immediately why the neighbor effect is an example of interactive behavioral contagion? (you mention yawn contagion, a non-interactive form, can also occur acoustically above)? I guess the assumption here is that the vocalizations involve >1 emitter? i.e., they suggest an ongoing social interaction between two conspecifics? Please clarify.

Response: We are sorry for not having been clearer. Indeed, the difference between non-interactive and interactive is between the type of behavior that is contagious: yawning and scratching are non-interactive, solitary behaviors, while affiliative and agonistic behaviors are interactive. The cited studies

on the neighbor effect, found evidence for social contagion of aggression and affiliation, i.e. interactive behavior. We clarified this in the revised version (line 48 – 49): *“A few studies on contagion of interactive (e.g. affiliative and aggressive) behavior in nonhuman primates [...]”*.

84-93: As they are currently worded and numbered, the predictions are hard to follow. I might suggest bullet points or some other format. I think it may also be important to allude to your sample demographics here (age-sex class of your subjects), and provide some further justification for some of the predictions. For instance, why would you expect social contagion to increase with the dominance rank of the target and the relationship between the target and observer? Could this be explained by a common mechanism – i.e., more attention towards the target?

Response: We thought about, but then refrained from presenting the predictions as a list of bullet points. We feel that it is in fact more readable when presented as one continuous paragraph with numbers delineating the separated predictions. Following the suggestion by the reviewer, we added the demographics of the sample, i.e. adult females, to the revised text (line 83). The justification for our prediction of contagion-enhancing effects of relationship quality and dominance rank are given in the previous paragraph and linked to increased emotional attachment or salience of signals from closely bonded or high status individuals (line 70 - 82): *“The degree of behavioral contagion of non-interactive behavior is often modulated by relationship quality between the stimulus individual and the bystander. Contagious yawning is enhanced when the triggering individual is a close social partner in geladas (Theropithecus gelada), bonobos (Pan paniscus) and humans (Demuru and Palagi 2012; for an opposing effect in contagious scratching see Laméris et al. 2020; Palagi et al. 2009; Palagi et al. 2014) or an in-group vs. outgroup individual in chimpanzees (Campbell and de Waal 2011). Yawn contagion is also facilitated when the triggering individual is of high social status, i.e. the dominant sex, in chimpanzees and bonobos (Demuru and Palagi 2012; Massen et al. 2012). Enhanced behavioral contagion as a function of stimulus’ dominance rank or relationship strength between the responder and the stimulus individual has been explained by increased emotional attachment and generally increased salience in signals of closely bonded or high status individuals (Deaner et al. 2005; Massen et al. 2012; Palagi et al. 2014; Schülke et al. 2020). Studies on behavioral contagion of interactive behavior have not tested the contagion modulation by relationship quality or dominance rank.”*

115: How was visual attention measured?

Response: we provide an explanation in the revised manuscript (line 126-127): *“The bystander female had to visually attend to the grooming interaction judged from head orientation and gaze direction.”*

116-127: I am a bit concerned about the compatibility between PG and MC samples, though I can appreciate the authors did put thought into this already, and it can be tricky to find an appropriate matched-control.

Response: As stated above and in the manuscript, the PG and MC sessions were matched as closely as possible both in terms of the social environment (by making sure that the same individuals were present in both conditions) and in terms of activity (by time matching in a highly structure captive routine). We thus are confident that the conditions are highly compatible.

- 116: Does matched for time of day ensure that animals were engaged in a similar activity – e.g.,

social activities may occur more/less around feeding or other routine/non-routine management procedures. What the focal doing at the start of the PG/MC observation seems like it may be a more important criteria than time of day.

Response: Please see response to the editor and to this reviewer's comment above. Given the highly structured day in a captive setting, time matching makes it very likely that the group was involved in the same activity in the two corresponding conditions. We have not explicitly tested this though.

- 119-120: Wouldn't it also be important to control for whether non-focal individuals in proximity were currently engaged in a grooming bout? Authors specify that all individuals present during the PG observation had to be within 2m of the focal in the MC observation, but not whether their activity was matched (were the non-focal individuals engaged in similar behavior, or explicitly *not* grooming?)

Response: In the matched control session, as a requirement non-focal individuals not involved in grooming (line 127 – 129) *“For every PG observation, on the next possible day and matched for time of day, we collected a corresponding matched control (MC) observation on the same focal female using the same protocol only in the absence of any grooming interactions.”*

- 120-121: I understand that it is important to control for the social environment but what about the non-social environment; again, the activity of the focal seems crucial here (if the focal was feeding during the MC observation, they would be less likely to begin a grooming bout).

Response: I reiterate here my response to comment from above (comment 116-127): As stated above and in the manuscript, the PG and MC sessions were matched as closely as possible both in terms of the social environment (by making sure that the same individuals were present in both conditions) and in terms of activity (by time matching in a highly structure captive routine). We thus are confident that the conditions are highly compatible

- 121-125: Having tried to conduct MCs like this myself, I feel like this would be very difficult if not impossible (assuming individuals are constantly moving around). Could the authors provide a bit more detail on how they actually managed ensure all of these proximity criteria were met simultaneously (to be clear, I am more impressed than questioning their integrity!)

Response: We agree that this kind of study design is difficult and clearly not suited for other settings, such as large groups or natural environments. In our case, group sizes were small, reducing the number of possible combinations of individuals in proximity, space was confined to 25m², and visibility was excellent. In cases when it was not possible to collect data on the same time of the next day, we rather moved to the second day after the PG observation always maintaining the time matching as our priority.

173-176: 3 grooming interactions in both PG and MC *respectively* or combined across the two samples? What was the mean number of grooming interactions for the remaining females in the sample?

Response: We apologize for not having been clearer in the manuscript. We excluded females that had less than 3 grooming interactions in either PG or MC, not combined. However, all females reached at least 3 grooming interactions in the PG condition. All six exclusions were based on less than three interactions in the matched control condition. We clarified this in the revised version and also give averages and ranges of number of grooming interactions in the remaining sample (line 198 – 203): *“Following Berthier and Semple (2018), we used only those females for which we had data of at least 3*

grooming interactions in both PG and MC, respectively, which led to the exclusion of 6 females for which we had enough (> 3 grooming interactions) in PG, yet only 0 - 2 grooming interactions in MC observations. This led to a resulting sample size for this prediction of 13 focal females with an average of 5 (range: 3-8) interactions in PG and an average of 4 (range: 3-6) interactions in MC observations."

235-240: This is largely redundant with the introduction; in general, I found the discussion rather long. Removing this level of detail on prior studies (perhaps actually adding this back up in the relevant area of the introduction to better set the rationale for the current approach) could help streamline the discussion

Response: We agree and deleted this section (lines 235-240, original manuscript)

262: Not sure what 'gated' means here

Response: We replaced "gated" with "shaped" (line 290)